# Current practice and barriers in the implementation of ultrasound-based assessment of muscle mass in Japan: A nationwide, web-based cross-sectional study

Keishi Nawata[1]ᵒ, Nobuto Nakanishi[2]ᵒ*, Shigeaki Inoue[2], Keibun Liu[3], Masafumi Nozoe[4], Yuko Ono[2], Isamu Yamada[2], Hajime Katsukawa[5], Joji Kotani[2]

1 Department of Rehabilitation, University Hospital of Occupational and Environmental Health, Yahata-nishi, Kitakyushu, Fukuoka, Japan, 2 Division of Disaster and Emergency Medicine, Department of Surgery Related, Kobe University Graduate School of Medicine, Chuo-ward, Kobe, Japan, 3 Critical Care Research Group, Faculty of Medicine, University of Queensland and The Prince Charles Hospital, Chermside, Brisbane, QLD, Australia, 4 Department of Physical Therapy, Faculty of Nursing and Rehabilitation, Konan Women's University, Higashinada-ku, Kobe, Japan, 5 Department of Scientific Research, Japanese Society for Early Mobilization, Chiyoda-ku, Tokyo, Japan

ᵒ These authors contributed equally to this work.
* nobuto_nakanishi@yahoo.co.jp

**Data Availability Statement:** We deposited all relevant data in the Figshare repository below. https://doi.org/10.6084/m9.figshare.20341449.v1.

## Abstract

Muscle mass is an important factor for surviving an illness. Ultrasound has gained increased attention as a muscle mass assessment method because of its noninvasiveness and portability. However, data on the frequency of ultrasound-based muscle mass assessment are limited, and there are some barriers to its implementation. Hence, a web-based cross-sectional survey was conducted on healthcare providers in Japan, which comprised four parts: 1) participant characteristics; 2) general muscle mass assessment; 3) ultrasound-based muscle mass assessment; and 4) the necessity of, interest in, and barriers to its implementation. Necessity and interest were assessed using an 11-point Likert scale, whereas barriers were assessed using a 5-point Likert scale, in which "Strongly agree" and "Agree" were counted for the analysis. Of the 1,058 responders, 1,026 participants, comprising 282 physicians, 489 physical therapists, 84 occupational therapists, 120 nurses, and 51 dieticians, were included in the analysis. In total, 93% of the participants were familiar with general muscle mass assessment, and 64% had conducted it. Ultrasound-based muscle mass assessment was performed by 21% of the participants. Necessity and interest scored 7 (6–8) and 8 (7–10), respectively for ultrasound-based muscle mass assessment. The barriers to its implementation included lack of relevant education (84%), limited staff (61%), and absence of fixed protocol (61%). Regardless of the necessity of and interest in ultrasound-based muscle mass assessment, it was only conducted by one-fifth of the healthcare providers, and the most important barrier to its implementation was lack of education.

**Funding:** This research was funded by a crowdfunding project entitled the Muscle Atrophy Zero Project, using the platform "Otsucle" <https://otsucle.jp/cf/project/2553.html >. This work was partially supported by JSPS KAKENHI Grant Number JP20K17899. There was no additional external funding received for this study. The funders had no role in study design, data collection and analysis, decision to publish, or preparation of the manuscript.

**Competing interests:** The authors have declared that no competing interests exist.

## Introduction

Surviving an illness can be influenced by muscle mass [1, 2]. The condition wherein the muscle mass of patients decreases is termed sarcopenia [3]. The European Working Group on Sarcopenia in Older People and the Asian Working Group for Sarcopenia have defined sarcopenia as the state of decreased muscle mass along with impaired physical function [4, 5]. Sarcopenia can be categorized into primary and secondary sarcopenia. While primary sarcopenia is caused by advanced age, which accelerates muscle wastage, secondary sarcopenia can have various causes, such as disease, immobilization, and insufficient nutrition [6]. Because sarcopenia is associated with poor prognosis [7], its prevention is important.

Muscle mass can be assessed using various methods [8]. Dual-energy X-ray absorptiometry (DEXA) or bioelectrical impedance analysis (BIA) is used for detecting low muscle mass in patients with sarcopenia [4, 5]. According to the guidelines of the Asian Working Group for Sarcopenia, the diagnosis is made based on cutoff values for decreased muscle mass, which differ depending on the assessment method (DEXA: cutoffs for women and men are 5.4 kg/m$^2$ and 7.0 kg/m$^2$, respectively; BIA: cutoffs for women and men are 5.7 kg/m$^2$ and 7.0 kg/m$^2$, respectively) [5]. However, there are numerous limitations associated with the use of DEXA and BIA. DEXA requires the transfer of patients to the examination room but involves low radiation exposure. Although BIA can be easily performed at the bedside, the measurement value is influenced by several factors, including posture, fever, fluid balance, and body mass index [9, 10]. Recently, ultrasound has gained attention as a muscle mass assessment method [11] because it can be used to assess muscle mass and sarcopenia [12, 13]. Therefore, the inclusion of ultrasound as a method to diagnose sarcopenia has been proposed [14, 15].

There are several advantages in the use of ultrasound for muscle mass assessment. Ultrasound is noninvasive and can be employed by various levels of healthcare providers [16]. It can be used continuously to monitor muscle mass because it is available at the bedside. Furthermore, unlike DEXA and BIA, ultrasound can visually assess skeletal muscle mass without being influenced by fluid balance [10]. However, the limitations of the technique include the high dependency on the examiner's skills. It has recently been demonstrated that ultrasound-based muscle mass assessment can be correctly performed by physicians, physical therapists, nurses, and dieticians if phantom training is given [17]. Despite the potential of ultrasound, data on the frequency of ultrasound-based muscle mass assessment are limited, and there are some barriers to its implementation.

It is hypothesized that some healthcare providers are interested in using ultrasound but do not do so because of certain barriers. This study aimed to identify the current practices and barriers in the implementation of ultrasound-based muscle mass assessment to facilitate the use of the technique for the proper management of sarcopenia.

## Materials and methods

### Study design and setting

This web-based, cross-sectional survey was conducted in Japan over a period of 1 month from August 1, 2021, to August 31, 2021. The target population of this survey included healthcare providers, such as physicians, nurses, physical therapists, occupational therapists, and dieticians, working in Japanese hospitals or clinics. Nonhealthcare providers were excluded. Those healthcare professions with insufficient (<50) participants were also excluded. The questionnaire was designed in such a way that the participants could not proceed unless each question is answered. This study complied with the principles of the Declaration of Helsinki and was approved by the clinical research ethics committees of Kobe University (approval number:

B210114). This study was registered in UMIN-CTR (UMIN 000044276). The requirement of written informed consent was waived because of the use of anonymized data. Informed consent was obtained via the online platform before participation. In the explanation form, the participants were requested to answer this questionnaire only once. Duplicate responses were identified by scrutinizing the sex, area, occupation, years of clinical experience, hospital type, and the number of hospital beds for similarity.

### Recruitment and distribution

This online survey was conducted using Google Forms. The participants were recruited using voluntary sampling. They were invited to complete the online survey via social media platforms, such as Facebook, Twitter, Instagram, and LinkedIn. This anonymous questionnaire was also distributed via email networks, such as Emergency Medicine Alliance, Japanese Society of Education for Physicians and Trainees in Intensive Care, JHospitalist Network, and Infectious Diseases Association for Teaching and Education in Nippon. Furthermore, this questionnaire was distributed among coauthors and individuals who supported the survey. No incentives were offered to the participants.

### Survey items

When selecting the survey items, we referred to previous studies that evaluated the use of ultrasound-based muscle mass assessment among physiotherapists [18, 19] or physicians in the rheumatological department [20]. The survey items were drafted by the survey team members (physicians and physiotherapists). During the planning phase, the validity and relevance of each item was verified by other healthcare providers, including nurses and nutritionists. The questionnaire is presented in the supplemental file (S1 Questionnaire).

This online survey included 31 items with four sections. The first section collected data on the participants' characteristics, including sex, years of clinical experience, occupation (physician, physical therapist, occupational therapist, nurse, dietician, or others); the area of residence; the type of hospital where they work; and the number of hospital beds.

The second section assessed the current practices in general muscle mass assessment, including DEXA, BIA, computed tomography, ultrasound, and limb circumference measurement. This section collected details on knowledge and practice. The knowledge section comprised two subsections that examined the number of participants familiar with muscle mass assessment and the known methods. The practice section comprised three subsections that examined the number of participants who conduct muscle mass assessment, the method used, the place where the assessment is conducted, and the purpose of the assessment.

The third section of this survey evaluated the current practice of ultrasound-based muscle mass assessment. Furthermore, this section collected details on the knowledge and practice of ultrasound-based muscle mass assessment. The knowledge section comprised two subsections that examined the number of participants familiar with ultrasound-based muscle mass assessment and how they learnt about this method. This question had multiple-choice answers. The practice section included questions on experience in using ultrasound, accessibility of the ultrasound device, existence of staff capable of conducting ultrasound-based muscle mass assessment, purpose of the assessment, learning methods, and muscles assessed.

The fourth section comprised barriers to and interest in conducting ultrasound-based muscle mass assessment. The barriers were assessed using a 5-point Likert scale, with the responses ranging from "Strongly disagree" to "Strongly agree." The barrier section comprised five parts, including 1) costs, such as cost of purchasing the equipment; 2) limited staffing or heavy workload; 3) insufficient education, such as absence of teaching staff or lectures; 4) non-reliability

of the assessment, such as insufficient evidence; and 5) lack of an organized protocol. For further analysis, the numbers of participants who answered "Agree" or "Strongly agree" were added to perform a comparison. The necessity of or interest in conducting ultrasound-based muscle mass assessment was assessed using an 11-point Likert scale, with scores ranging from 0 to 10. Finally, interest in participation in lectures or practical seminars on ultrasound-based muscle mass assessment as well as in utilizing such opportunities was evaluated.

### Primary and secondary analyses

This study aimed to identify current practices in and barriers to implementing ultrasound-based muscle mass assessment. The primary analysis was to conduct descriptive analyses among all healthcare workers with sufficient population where the number of participants was ≥50. The secondary analysis was to assess the differences among occupations and the differences between participants with and without experience in ultrasound-based muscle mass assessment.

### Statistical analyses

Data were downloaded from the filled Google Forms into Microsoft Excel and subsequently transferred into the JMP statistical software version 13.1.0 (SAS Institute Inc., Cary, NC, USA) for statistical analysis. The data were presented in the form of frequencies, percentages, means ± standard deviations, and medians (interquartile ranges). The Student's t-test or Mann–Whitney U test was used to calculate the mean or median difference in the variables, respectively. The chi-squared test and Fisher's exact test were used to determine the associations between categorical variables. For multiple comparisons of the difference among the five occupations, the data were further analyzed using Bonferroni correction (significant at $p < 0.01$). Participants with duplicate answers or missing data were excluded. The sample size was not calculated *a priori* because of the exploratory nature of this study. The level of significance was set at 0.05.

## Results

Overall, 1,058 individuals across Japan responded to the survey (S1 Fig and S1 Table). The trend of the total responses during the survey period is summarized in S2 Fig. Of the 1,058 responders, we excluded 1 nonhealthcare provider and 31 insufficient group populations, including 13 pharmacists, 7 clinical engineers, 5 speech therapists, and 6 other occupations (dentist, medical technologist, biological researcher, practitioner in acupuncture and moxibustion, bonesetter, and certified social worker). There were no duplicate or missing responses. Finally, 1,026 participants were considered for further analysis (Fig 1). The occupations included were 282 physicians, 489 physical therapists, 84 occupational therapists, 120 nurses, and 51 dieticians. A total of 741 (72%) participants were men, and the average clinical experience of the participants was 11 (6–17) years (Table 1). The participants' physical characteristics are summarized in Table 1.

Totally, 93% of the participants were familiar with general muscle mass assessments. Specifically, 99% of physical therapists and 98% of dieticians were familiar with any type of muscle mass assessment. Limb circumference measurement was the most well-known method (86%), followed by BIA (61%) and computed tomography (60%). Muscle mass assessment was conducted by 64% of the participants; 85% of physical therapists and 86% of dieticians used the assessment. Limb circumference measurement was the most commonly used method (53%), followed by BIA (31%) and ultrasound (14%). BIA was most commonly used by dieticians (61%), whereas ultrasound was most commonly used by physical therapists (22%). Muscle

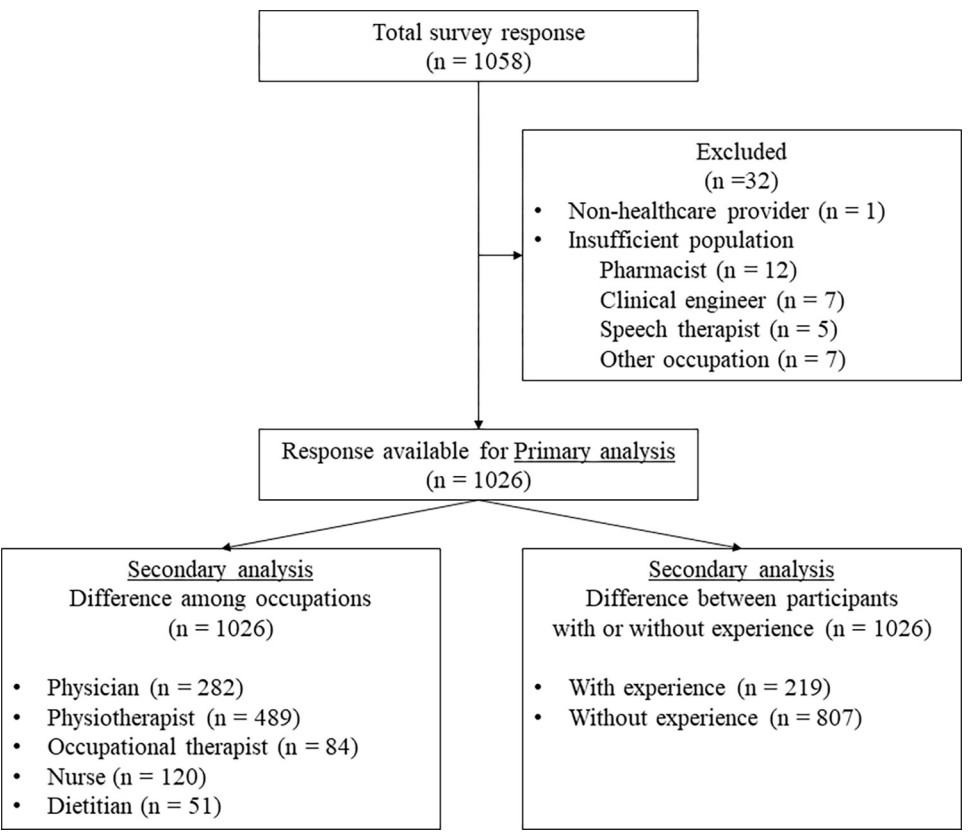

**Fig 1. Flow of participants through the study.** Among 1,058 respondents, 1,026 were included in the primary analysis. Secondary outcomes included the differences based on occupation and the differences in ultrasound-based muscle mass assessment between experienced and inexperienced participants.

mass assessment was frequently conducted at the general ward (48%) and for outpatients (20%). The purposes of assessing muscle mass were clinical evaluation (60%) and research application (29%).

A total of 53% of the participants had used ultrasound, with physicians accounting for the majority at 88% (Table 2). Likewise, access to ultrasound was the greatest for physicians at 94%. Ultrasound-based muscle mass assessment was known by 51% and conducted by 21% of the participants. The most common method of learning about ultrasound-based muscle mass assessment was reading academic papers (34%). The purpose of assessment was clinical evaluation in 20% and research application in 16%. The common muscle properties evaluated were quadriceps femoris muscle thickness (16%) and rectus femoris muscle cross-sectional area (10%).

The most common barrier was the lack of education (84%), followed by limited staffing (61%) and lack of an established protocol (61%) (Fig 2 and S2 Table). These barriers were perceived differently between ultrasound-based assessments by experienced and inexperienced participants (Table 3 and S3 Table). Lack of education was more frequently considered a barrier by inexperienced participants, whereas cost and reliability were more frequently considered as barriers by experienced participants. The necessity of and interest in ultrasound-based muscle mass assessment scored 7 (6–8) and 8 (7–10), respectively (Fig 3). In terms of interest, the muscle that received the most interest was the thigh muscle (68%), followed by the lower leg muscle (49%) and the diaphragm (46%). Although 77% of the participants were interested in attending a lecture or hands-on seminar, only 15% had actually participated in them.

**Table 1. Background and general muscle mass assessment.**

| Variables | Overall | Physician | Physical therapist | Occupational therapist | Nurse | Dietician |
|---|---|---|---|---|---|---|
| | | | Difference in each occupation | | | |
| | (n = 1026) | (n = 282) | (n = 489) | (n = 84) | (n = 120) | (n = 51) |
| Sex (Men) | 741 (72) | 236 (84) | 398 (81) | 50 (60) | 50 (42) | 7 (14) |
| Clinical experience, years | 11 (6–17) | 12 (7–20) | 10 (5–15) | 11 (4–16) | 15 (10–20) | 6 (2–16) |
| Type of hospital | | | | | | |
| University hospital | 269 (26) | 85 (30) | 101 (21) | 19 (23) | 45 (38) | 19 (37) |
| Municipal hospital | 713 (70) | 195 (69) | 357 (73) | 58 (69) | 74 (62) | 29 (57) |
| Number of hospital beds | 400 (190–655) | 500 (300–716) | 328 (152–600) | 284 (123–615) | 500 (267–734) | 464 (155–827) |
| Participants who know muscle mass assessment | 952 (93) | 246 (87) | 486 (99) | 77 (92) | 93 (78) | 50 (98) |
| The assessment method which participants know | | | | | | |
| Dual-energy X-ray absorptiometry | 391 (38) | 61 (22) | 265 (54) | 15 (18) | 8 (7) | 42 (82) |
| Bioelectrical impedance analysis | 624 (61) | 129 (46) | 372 (76) | 45 (54) | 29 (24) | 49 (96) |
| Computed tomography | 611 (60) | 157 (56) | 341 (70) | 38 (45) | 42 (35) | 33 (65) |
| Ultrasound | 585 (57) | 99 (35) | 370 (76) | 46 (55) | 48 (40) | 22 (43) |
| Limb circumference | 878 (86) | 211 (75) | 474 (97) | 75 (89) | 78 (65) | 40 (78) |
| Participants who conduct muscle mass assessment | 658 (64) | 110 (39) | 416 (85) | 55 (66) | 33 (28) | 44 (86) |
| The assessment method which participants conduct | | | | | | |
| Dual-energy X-ray absorptiometry | 38 (4) | 8 (3) | 22 (5) | 0 (0) | 3 (3) | 5 (10) |
| Bioelectrical impedance analysis | 313 (31) | 46 (16) | 197 (40) | 25 (30) | 14 (12) | 31 (61) |
| Computed tomography | 46 (5) | 18 (6) | 20 (4) | 2 (2) | 3 (3) | 3 (6) |
| Ultrasound | 139 (14) | 23 (8) | 107 (22) | 2 (2) | 4 (3) | 3 (6) |
| Limb circumference | 546 (53) | 75 (27) | 371 (76) | 56 (67) | 21 (18) | 23 (45) |
| The place of assessment | | | | | | |
| Outpatients | 204 (20) | 55 (20) | 106 (22) | 11 (13) | 7 (6) | 25 (49) |
| Emergency room | 22 (2) | 3 (1) | 10 (2) | 2 (2) | 4 (3) | 3 (6) |
| Intensive care unit | 110 (11) | 17 (6) | 66 (14) | 3 (4) | 18 (15) | 6 (12) |
| Acute care unit | 104 (10) | 10 (4) | 69 (14) | 8 (10) | 11 (9) | 6 (12) |
| General ward | 496 (48) | 61 (22) | 340 (70) | 47 (56) | 14 (12) | 34 (67) |
| The purpose of assessment | | | | | | |
| Clinical evaluation | 613 (60) | 89 (32) | 403 (82) | 53 (63) | 26 (22) | 42 (82) |
| Research application | 297 (29) | 49 (17) | 193 (40) | 16 (19) | 16 (13) | 23 (45) |

Data were presented as number (percentage) or median (interquartile range) unless otherwise indicated.

## Discussion

In this nationwide questionnaire survey that included 1,026 participants, the current practices in and barriers to implementing ultrasound-based muscle mass assessment were investigated. The main finding of this study was that 21% of the healthcare providers who responded to the survey had conducted ultrasound-based muscle mass assessments. The most significant barrier to its implementation was lack of education. Although the necessity of and interest in muscle mass assessment were high, there was not enough opportunity to learn the technique. These findings highlight the importance of educating the healthcare providers on ultrasound-based muscle mass assessment.

Unlike other muscle mass assessment methods, the accuracy of ultrasound-based assessment depends highly on the examiner's skill. The procedure, including probe compression and measurement angle, can lead to approximately 5%–10% difference in the measured value [17]. As less than half of all healthcare providers, except physicians, had not used ultrasound

**Table 2. Ultrasound-based muscle mass assessment.**

| Variables | Overall | Physician | Physical therapist | Occupational therapist | Nurse | Dietician |
|---|---|---|---|---|---|---|
| | | | | Difference in each occupation | | |
| | (n = 1026) | (n = 282) | (n = 489) | (n = 84) | (n = 120) | (n = 51) |
| Participants who have used ultrasound | 548 (53) | 247 (88) | 229 (47) | 16 (19) | 44 (37) | 12 (24) |
| Participants who have access to ultrasound in their facility | 718 (70) | 264 (94) | 305 (62) | 34 (41) | 93 (78) | 22 (43) |
| Participants who know ultrasound-based assessment | 521 (51) | 79 (28) | 342 (70) | 35 (42) | 43 (36) | 22 (43) |
| The process to have known ultrasound-based assessment | | | | | | |
| Academic conference | 346 (34) | 31 (11) | 252 (52) | 17 (20) | 34 (28) | 12 (24) |
| Lecture or seminar | 308 (30) | 31 (11) | 222 (45) | 22 (26) | 21 (18) | 12 (24) |
| Academic paper | 352 (34) | 45 (16) | 251 (51) | 25 (30) | 21 (18) | 10 (20) |
| Book | 168 (16) | 18 (6) | 123 (25) | 9 (11) | 14 (12) | 4 (8) |
| Social Networking Service | 107 (10) | 25 (9) | 60 (12) | 5 (6) | 14 (12) | 3 (6) |
| Participants who have conducted ultrasound-based assessment | 219 (21) | 42 (15) | 162 (33) | 7 (8) | 4 (3) | 4 (8) |
| Participants who have colleague to conduct ultrasound-based assessment | 235 (23) | 34 (12) | 162 (33) | 18 (21) | 14 (13) | 6 (12) |
| The purpose of ultrasound-based assessment | | | | | | |
| Clinical evaluation | 208 (20) | 43 (15) | 147 (30) | 10 (12) | 5 (4) | 3 (6) |
| Research application | 167 (16) | 31 (11) | 121 (25) | 5 (6) | 6 (5) | 4 (8) |
| The method to study ultrasound-based assessment | | | | | | |
| Academic conference | 145 (14) | 17 (6) | 113 (23) | 5 (6) | 6 (5) | 4 (8) |
| Lecture or seminar | 180 (18) | 19 (7) | 134 (27) | 12 (14) | 10 (8) | 5 (10) |
| Academic paper | 212 (21) | 34 (12) | 155 (32) | 11 (13) | 9 (8) | 3 (6) |
| Book | 136 (13) | 20 (7) | 104 (21) | 6 (7) | 5 (4) | 1 (2) |
| Social Networking Service | 38 (4) | 7 (3) | 24 (5) | 2 (2) | 4 (3) | 1 (2) |
| Muscle assessed by ultrasound | | | | | | |
| Biceps brachii muscle thickness | 26 (3) | 5 (2) | 16 (3) | 2 (2) | 3 (3) | 0 (0) |
| Biceps brachii muscle cross-sectional area | 12 (1) | 5 (2) | 6 (1) | 0 (0) | 1 (1) | 0 (0) |
| Quadriceps femoris muscle thickness | 168 (16) | 26 (9) | 131 (27) | 6 (7) | 2 (2) | 3 (6) |
| Rectus femoris muscle cross-sectional area | 102 (10) | 16 (6) | 79 (16) | 2 (2) | 4 (3) | 1 (2) |
| Lower leg muscle thickness | 51 (5) | 9 (3) | 39 (8) | 0 (0) | 3 (3) | 0 (0) |
| Lower leg muscle cross-sectional area | 39 (4) | 6 (2) | 29 (6) | 0 (0) | 3 (3) | 1 (2) |
| Diaphragm | 39 (4) | 11 (4) | 27 (6) | 0 (0) | 1 (1) | 0 (0) |
| Muscle interested to be assessed in ultrasound-based assessment | | | | | | |
| Upper limb | 314 (31) | 91 (32) | 114 (23) | 52 (62) | 37 (31) | 20 (39) |
| Thigh muscle | 697 (68) | 172 (61) | 376 (77) | 34 (41) | 82 (68) | 33 (65) |
| Lower leg muscle | 499 (49) | 102 (36) | 287 (59) | 29 (35) | 57 (48) | 24 (47) |
| Diaphragm | 476 (46) | 117 (42) | 254 (52) | 25 (30) | 67 (56) | 13 (26) |
| Participants who have joined lecture or hands-on seminar | 158 (15) | 20 (7) | 118 (24) | 9 (11) | 5 (4) | 6 (12) |
| Participants who want to join lecture or hands-on seminar | 788 (77) | 212 (75) | 394 (81) | 53 (63) | 95 (79) | 34 (67) |
| The style to join | | | | | | |
| Lecture | 824 (80) | 229 (81) | 411 (84) | 58 (69) | 89 (74) | 37 (73) |
| Hands-on seminar | 804 (78) | 223 (79) | 396 (81) | 59 (70) | 91 (76) | 35 (69) |

Data were presented as number (percentage) unless otherwise indicated.

assessment, proper training is essential. Educating healthcare providers at different levels can improve their skills in conducting this assessment [21]. In this questionnaire survey, 70%–80% of the participants were keen on participating in any educational opportunity, whereas merely 15% had actually participated in such an opportunity. The most common learning method

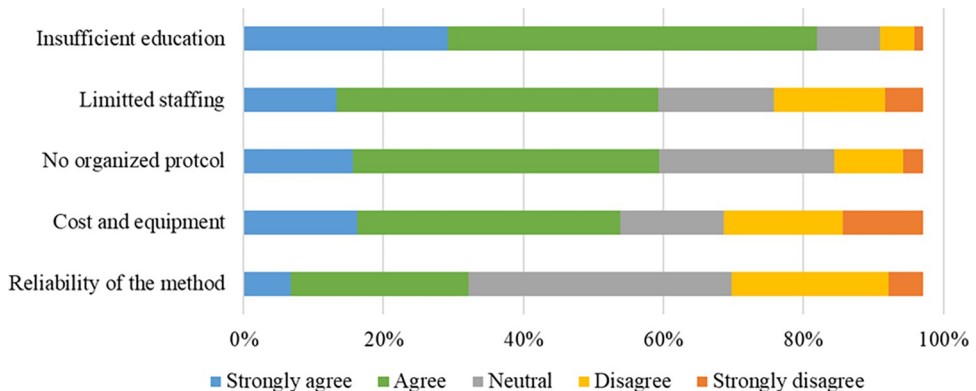

**Fig 2. The barriers for implementing ultrasound-based muscle mass assessment.** The barriers were assessed using a 5-point Likert scale with "Strongly agree" and "Agree," which were used for the comparison.

was limited to reading academic papers, which accounted for 21% of the participants. Consistent with our findings, other surveys on physiotherapists or rheumatologists have also indicated that one of the most common barriers is the lack of training [18–20]. According to Ellis et al., 76% of non-ultrasound users answered that lack of training was the barrier to using the method in the international survey [18]. Potter et al. reported that this barrier accounted for 63% of the participants in the United Kingdom [19]. The educational barrier was much stronger among inexperienced participants than among experienced ones, which suggests the difficulty in learning the technique because of insufficient educational opportunities.

Moreover, there are numerous other barriers to overcome. A lack of established protocol was also perceived as a barrier; there is no fixed protocol to assess muscle mass using ultrasound [8]. Regarding the commonly assessed femur muscles, measuring the cross-sectional area of the rectus femoris is more useful than measuring its thickness because the latter does not reflect physical function [22]. However, the thickness was more commonly measured, partly because of the lack of protocol. Contrary to expectations, approximately 70% of the participants had access to ultrasound equipment, thus indicating that practitioners in Japan can access ultrasound relatively easily. Although access to the equipment acted as the barrier in 56% of non-ultrasound users in a previous study [18], it may vary from country to country. Indeed, it is a serious problem to ensure ultrasound equipment and its maintenance in developing countries [23]. In our study, costs, such as those of purchasing the equipment, were much less of an obstacle for inexperienced examiners, reflecting the accessibility of the ultrasound. However, accessibility also varies among the occupations, with 41% of occupational therapists, 43% of dieticians, and 90% of physicians having access to ultrasound. The reliability

**Table 3. Differences in ultrasound-based assessment between experienced and inexperienced participants.**

| Variables | Overall | Experienced | Inexperienced | p value |
|---|---|---|---|---|
| | (n = 1026) | (n = 219) | (n = 807) | |
| Barriers (Agree or Strongly agree) | | | | |
| Education | 866 (84) | 174 (80) | 692 (86) | 0.02 |
| Limited staffing | 626 (61) | 142 (65) | 484 (60) | 0.19 |
| No organized protocol | 628 (61) | 127 (58) | 501 (62) | 0.27 |
| Cost of the equipment | 569 (56) | 159 (73) | 410 (51) | < 0.01 |
| Reliability of the assessment | 340 (33) | 88 (40) | 252 (31) | 0.01 |

The number of participants who answered "Agree" or "Strongly agree" was counted, and the data were presented as number (percentage).

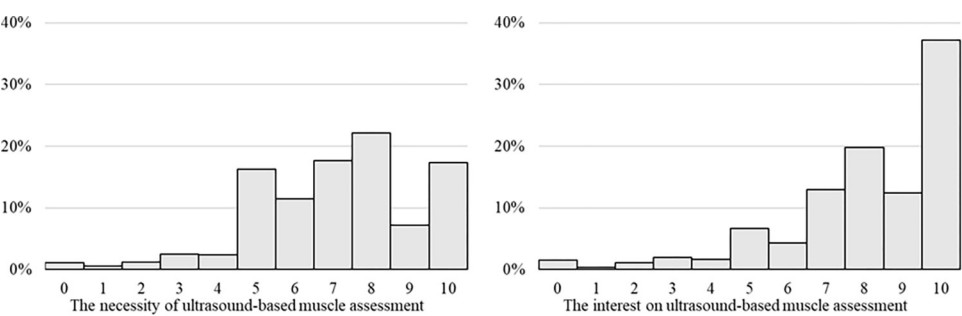

**Fig 3. The necessity of and interest in ultrasound-based muscle mass assessment.** The necessity of and interest in ultrasound-based muscle mass assessment were assessed using an 11-point Likert scale ranging from 0 to 10. The necessity of and interest in ultrasound-based muscle mass assessment were 7 (6–8) and 8 (7–10), respectively.

of ultrasound measurement was more frequently considered a barrier by experienced examiners, probably because they were aware of the difficulty in achieving reproducibility. In case of uncertain reliability, an ultrasound phantom can be used to confirm the reliability of the ultrasound measurement [17].

As per the findings of this survey, the most common method of assessing muscle mass was not ultrasound but limb circumference measurement, which was performed by approximately half of the participants. Although limb circumference can be measured in any country without any specific equipment [24], we need to know limb circumference measurement does not have a good discriminatory power for identifying low skeletal muscle mass because the indirect muscle mass assessment is affected by numerous factors [25]. Similarly, BIA is an indirect method of assessing muscle mass and is influenced by fluid balance [10]. Although approximately 60% of the participants were familiar with BIA and ultrasound, ultrasound-based assessment methods were conducted only in one-half of BIA. Furthermore, ultrasound-based muscle mass assessment was performed mostly in the general ward and not for patients in acute stages, including those admitted to the intensive care unit (ICU). Among those in the ICU, muscle mass is important for surviving an illness; muscle atrophy is a serious problem, wherein approximately 15%–20% of muscle mass is wasted during the first week of the ICU stay [26]. Therefore, muscle mass assessment in the acute stages is also important.

Notably, the responses were different among the participants from different occupations. Ultrasound-based muscle mass assessment was most frequently conducted by physical therapists (33%), which is consistent with a previous study reporting that 38% of physiotherapists use ultrasound [18]. Moreover, 15% of physicians performed ultrasound-based muscle mass assessment, which is also consistent with a previous study reporting a finding of 17% [20]. Ultrasound is an important tool for assessing physical function. Most occupational therapists were interested in ultrasound-based upper limb assessment, which has been proven to reflect whole-body muscle mass [12]. Furthermore, 60% of the dieticians used BIA, whereas only 6% used ultrasound for muscle mass assessment. Ultrasound is a promising tool for dieticians because it helps assess the patient's nutritional status [27, 28]. As previously discussed, ultrasound can be used by healthcare providers involved in various levels of care as long as they are sufficiently trained. Therefore, training is essential for most healthcare providers to ensure that they can accurately assess muscle mass using ultrasound.

## Limitations

First, this survey was conducted using snowballing recruitment via social networking service and a mailing list. Therefore, the response rate could not be determined. Furthermore, there is

a possibility of bias toward those participants interested in ultrasound-based muscle mass assessment; therefore, this survey does not completely represent the entire population of healthcare providers. However, as illustrated in S1 Fig, the participants of this survey were from across the country, thus reflecting the trend of ultrasound-based muscle mass assessment in Japan. Second, this study was conducted in Japan; However, some questionnaire responses were consistent with previous surveys, possibly presenting important suggestions in other countries. Finally, the participants were requested to respond just once, and duplicate responses were discarded; however, there was no systematic prevention of such responses.

## Conclusions

This questionnaire study examined the current practices in and barriers to implementing ultrasound-based muscle mass assessment. The practice of using this method varied widely among different occupations. Although the necessity of and interest in ultrasound-based assessment was high among most occupations, only 21% of the respondents had used the procedure. Lack of education was identified to be the most important barrier that must be overcome to widely implement ultrasound-based assessment.

## Supporting information

**S1 Fig. The response distribution in Japan.** Response was obtained across Japan. Response area is colored in this map. This figure was created using R statistical software (version 4.1.0).
(DOCX)

**S2 Fig. The trend of response in survey period.** Response was obtained as shown in the figure. The mailing lists used are shown in different colored arrows.
(DOCX)

**S1 Table. Response distribution among different occupations.**
(DOCX)

**S2 Table. Differences in barriers to and interest in ultrasound-based assessment among different occupations.**
(DOCX)

**S3 Table. Difference between participants with or without ultrasound-based muscle mass assessment.**
(DOCX)

**S1 Questionnaire. Questionnaire for assessing current status regarding ultrasound-based muscle mass assessment.**
(DOCX)

## Acknowledgments

The authors thank Takeshi Unoki (Department of Acute and Critical Care Nursing, School of Nursing, Sapporo City University), Kazuki Okura (Department of Rehabilitation, Akita University Hospital), Masatsugu Okamura (Department of Rehabilitation, Yokohama City University Hospital), Ayato Shinohara (Department of Rehabilitation, Fujita Health University Hospital), Kohei Tanaka (Department of Rehabilitation Medicine, Osaka Police Hospital), and Sho Katayama (Department of Rehabilitation Medicine, Okayama University Hospital) for their cooperation and support during the study. The authors also thank people who supported

the nonprofit crowdfunding of Muscle Atrophy Zero Project, which aims to prevent muscle atrophy in critically ill patients.

## Author Contributions

**Conceptualization:** Keishi Nawata, Nobuto Nakanishi, Keibun Liu, Masafumi Nozoe.

**Data curation:** Keishi Nawata, Nobuto Nakanishi, Masafumi Nozoe.

**Formal analysis:** Nobuto Nakanishi, Keibun Liu.

**Funding acquisition:** Nobuto Nakanishi, Shigeaki Inoue, Joji Kotani.

**Investigation:** Keishi Nawata, Nobuto Nakanishi.

**Methodology:** Keibun Liu, Masafumi Nozoe, Yuko Ono, Hajime Katsukawa.

**Project administration:** Nobuto Nakanishi.

**Supervision:** Shigeaki Inoue, Isamu Yamada, Joji Kotani.

**Writing – original draft:** Keishi Nawata, Nobuto Nakanishi.

**Writing – review & editing:** Nobuto Nakanishi, Shigeaki Inoue, Yuko Ono, Hajime Katsukawa.

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
