## [Decision Letter · Decision Letter 0]

20 Jun 2022

PONE-D-21-34367Current Practice and Barriers for Implementing Ultrasound-Based Muscle Mass Assessment in Japan: A Nationwide, Web-Based Cross-Sectional StudyPLOS ONE

Dear Dr. Nakanishi,

Thank you for submitting your manuscript to PLOS ONE. Firstly, we would like to apologize for the delay in processing your manuscript. It has been exceptionally difficult to secure reviewers to evaluate your study. We have now received one completed review, which is available below. The reviewer has raised significant scientific concerns about the study that need to be addressed in a revision.

Please note that we have only been able to secure a single reviewer to assess your manuscript. We are issuing a decision on your manuscript at this point to prevent further delays in the evaluation of your manuscript. Please be aware that the editor who handles your revised manuscript might find it necessary to invite additional reviewers to assess this work once the revised manuscript is submitted. However, we will aim to proceed on the basis of this single review if possible.

After careful consideration, we feel that it has merit but does not fully meet PLOS ONE’s publication criteria as it currently stands. Therefore, we invite you to submit a revised version of the manuscript that addresses the points raised during the review process.

We look forward to receiving your revised manuscript.

Kind regards,

Miquel Vall-llosera Camps

Senior Editor

PLOS ONE

Journal Requirements:

A clean copy of the edited manuscript (uploaded as the new *manuscript* file).

3. Thank you for stating in your Funding Statement: "This research was funded by a crowdfunding project entitled the Muscle Atrophy Zero Project, using the platform “Otsucle” <https://otsucle.jp/cf/project/2553.html  >. This work was partially supported by JSPS KAKENHI Grant Number JP20K17899."

4. Thank you for stating the following financial disclosure: "This research was funded by a crowdfunding project entitled the Muscle Atrophy Zero Project, using the platform “Otsucle” <https://otsucle.jp/cf/project/2553.html  >. This work was partially supported by JSPS KAKENHI Grant Number JP20K17899."

Please state what role the funders took in the study.  If the funders had no role, please state: "The funders had no role in study design, data collection and analysis, decision to publish, or preparation of the manuscript.

5. Thank you for stating the following in the Acknowledgments Section of your manuscript: "The authors thank Takeshi Unoki (Department of Acute and Critical Care Nursing, School of Nursing, Sapporo City University),  Kazuki Okura (Department of Rehabilitation, Akita University Hospital), Masatsugu Okamura (Department of Rehabilitation, Yokohama City University Hospital), Ayato Shinohara (Department of Rehabilitation, Fujita Health University Hospital), Kohei Tanaka (Department of Rehabilitation Medicine, Osaka Police Hospital), Sho Katayama (Department of Rehabilitation Medicine, Okayama University Hospital) for their cooperation and support during the study, and people who supported the nonprofit crowdfunding of Muscle Atrophy Zero Project, which aims to prevent muscle atrophy in critically ill patients. This work was partially supported by JSPS KAKENHI Grant Number JP20K17899."

Please remove any funding-related text from the manuscript and let us know how you would like to update your Funding Statement. Currently, your Funding Statement reads as follows: "This research was funded by a crowdfunding project entitled the Muscle Atrophy Zero Project, using the platform “Otsucle” <https://otsucle.jp/cf/project/2553.html  >. This work was partially supported by JSPS KAKENHI Grant Number JP20K17899."

6. We note that you have indicated that data from this study are available upon request. PLOS only allows data to be available upon request if there are legal or ethical restrictions on sharing data publicly. For more information on unacceptable data access restrictions, please see http://journals.plos.org/plosone/s/data-availability#loc-unacceptable-data-access-restrictions. 

7. Your abstract cannot contain citations. Please only include citations in the body text of the manuscript, and ensure that they remain in ascending numerical order on first mention.

8. We note that Figure S2 in your submission contain map images which may be copyrighted. All PLOS content is published under the Creative Commons Attribution License (CC BY 4.0), which means that the manuscript, images, and Supporting Information files will be freely available online, and any third party is permitted to access, download, copy, distribute, and use these materials in any way, even commercially, with proper attribution. For these reasons, we cannot publish previously copyrighted maps or satellite images created using proprietary data, such as Google software (Google Maps, Street View, and Earth). For more information, see our copyright guidelines: http://journals.plos.org/plosone/s/licenses-and-copyright.

a. You may seek permission from the original copyright holder of Figure S2 to publish the content specifically under the CC BY 4.0 license.  

Reviewers' comments:

Reviewer's Responses to Questions

**Comments to the Author**

1. Is the manuscript technically sound, and do the data support the conclusions?

Reviewer #1: Yes

2. Has the statistical analysis been performed appropriately and rigorously? 

Reviewer #1: No

3. Have the authors made all data underlying the findings in their manuscript fully available?

Reviewer #1: No

4. Is the manuscript presented in an intelligible fashion and written in standard English?

Reviewer #1: No

5. Review Comments to the Author

Reviewer #1: Sarcopenia is a clinical condition characterized by progressive and generalized loss of muscle mass and muscular force, accompanied by an elevated risk of adverse events. The diagnostis criteria require assessment of muscle mass, muscle force and muscle function impairment, all these exams are time-consuming, with limited use in everyday clinical practice. On the other hand ultrasound measurement of muscle mass impairment can be used as a quick screening

test to assess the presence of sarcopenia in elderly patients.

The reviewed paper is interesting, Authors analysed barriers against wider use of this method in clinical practice, trying to identify motivations that prevent healthcare professional to use it correctly.

Introduction:

Overall quality of the text is good, the language is clear and direct, but I would suggest to consider professional editing to remove some grammar issues. This would improve the quality of the work.

For example: page 2, line 55: “The state of decreased muscle mass in termed…” is not correct.

For example: page 3, line 63: “Muscle mass can be assessed by various measures..” the word “measures” is not the exact term that should be used, I would suggest to replace it with “methods”.

It would be also useful to describe in more detailed way the diagnostic criteria of sarcopenia, tht are actually in use.

Materials and Methods:

The study sample is quite good (1058 respondes) but is not homogeneous in relation to their professionality (physitians, nurses, etc..).

Only the descriptive analysis was realized, it would be interesting to examine the association between the groups.

For example, as reported in page 5, line 150: “. For the trend analysis, healthcare providers were included, where the number of participants was ≥50”. According to me the aim od the paper was s not to analyse the trend, I would suggest to specify what was the primary aim (from my point of view it was the descriptive analysis, the analysis of differences between groups (experienced vs. non-experienced).

According to me the analysis of the difference between healthcare professionales that can use tultrasound in their daily practice (physitian, physiotherapists, ets) and those that cannot use it (pharmacists, engineers, dentists) in not appropriate.

I would suggest exluding of these small sub-groups from the analysis as they are not supposed to use the analysed method in their practice.

The questionnaire should be provided, too.

Results:

The results are expressed as frequencies, the tables should report also the results of Likert scales.

The paper is interestig, but it reports only descripitive analysis of the healthcare professional’s opinions.

I would suggest to revise the paper and to submit it after again, trying to improve the language and to add more results, if possible.

6. PLOS authors have the option to publish the peer review history of their article (what does this mean?). If published, this will include your full peer review and any attached files.

Reviewer #1: No

---

## [Author Response · Author response to Decision Letter 0]

19 Jul 2022

Responses to Reviewer #1:

1. Sarcopenia is a clinical condition characterized by progressive and generalized loss of muscle mass and muscular force, accompanied by an elevated risk of adverse events. The diagnostic criteria require assessment of muscle mass, muscle force and muscle function impairment, all these exams are time-consuming, with limited use in everyday clinical practice. On the other hand, ultrasound measurement of muscle mass impairment can be used as a quick screening test to assess the presence of sarcopenia in elderly patients. The reviewed paper is interesting, Authors analyzed barriers against wider use of this method in clinical practice, trying to identify motivations that prevent healthcare professional to use it correctly.

a. We appreciate reviewer’s comments. We have been encouraged by reviewer’s positive comments.

2. Introduction:

Overall quality of the text is good, the language is clear and direct, but I would suggest to consider professional editing to remove some grammar issues. This would improve the quality of the work.

For example: page 2, line 55: “The state of decreased muscle mass in termed…” is not correct.

For example: page 3, line 63: “Muscle mass can be assessed by various measures.” the word “measures” is not the exact term that should be used, I would suggest to replace it with “methods”.?

a. We conducted a professional editing by ENAGO. We attached the proof of editing. We hope our revised manuscript meets your expectations.

3. It would be also useful to describe in more detailed way the diagnostic criteria of sarcopenia, that are actually in use.

a. We added more detailed description about the diagnostic criteria of sarcopenia in the background as following. 

According to the guidelines of Asian Working Group for Sarcopenia, the diagnosis of sarcopenia is made based on cutoff values for decreased muscle mass, which differ depending on the method used for assessment (DEXA: cutoff for women and men is 5.4 kg/m2 and 7.0 kg/m2, respectively; BIA: cutoff for women and men is 5.7 kg/m2 and 7.0 kg/m2, respectively).[5]

5. Chen LK, Woo J, Assantachai P, Auyeung TW, Chou MY, Iijima K, et al. Asian Working Group for Sarcopenia: 2019 consensus update on sarcopenia diagnosis and treatment. J Am Med Dir Assoc. 2020;21(3):300-7.e2. doi: 10.1016/j.jamda.2019.12.012. PMID: 32033882

4. Materials and Methods:

The study sample is quite good (1058 responders) but is not homogeneous in relation to their professionality (physicians, nurses, etc..).

Only the descriptive analysis was realized, it would be interesting to examine the association between the groups.

For example, as reported in page 5, line 150: “. For the trend analysis, healthcare providers were included, where the number of participants was ≥50”. According to me the aim of the paper was not to analyze the trend, I would suggest to specify what was the primary aim (from my point of view it was the descriptive analysis, the analysis of differences between groups (experienced vs. non-experienced).

a. Thank you for the opportunity to clarify this point. We added the description to clearly show our primary aim that our paper intended to identify current practices and barriers to implementing ultrasound-based muscle mass assessment. Primary analysis was descriptive analysis. Then, the secondary analyses were to seek the difference between experienced and unexperienced. Furthermore, we added the statistical analysis to examine the difference between the groups.

Primary and secondary analyses

The purpose of this study was to identify current practices and barriers to implementing ultrasound-based muscle mass assessment. The primary analysis was to conduct descriptive analyses among all healthcare workers with sufficient population where the number of participants was ≥50. The secondary analysis was to assess the differences among occupations and between participants with experience of ultrasound-based muscle mass assessment and those without it.

5. Materials and Methods:

According to me the analysis of the difference between healthcare professionals that can use ultrasound in their daily practice (physitian, physiotherapists, ets) and those that cannot use it (pharmacists, engineers, dentists) is not appropriate.

I would suggest excluding of these small sub-groups from the analysis as they are not supposed to use the analyzed method in their practice.?

a. We excluded the small sub-groups (pharmacists, engineers, dentists…) from our analysis. We revised the main Figure 1 (Patient flow) in which pharmacists (n = 12), clinical engineer (n = 7), speech therapist (n = 5), and other occupation (n = 7) were excluded. Furthermore, we reanalyzed secondary analysis about difference between participants with or without experience in the total 1026 participants. We totally revised our manuscript regarding the point throughout our manuscript.

Method

We also excluded those healthcare professions with insufficient (<50) participants.

Results

Overall, 1,058 individuals responded to the survey from all across Japan (S1 Fig. and S1 Table). The trend of the total responses during the survey period is summarized in S2 Figure. Of the 1,058 responders, we excluded 1 nonhealthcare provider and 31 insufficient group populations, including 13 pharmacists, 7 clinical engineers, 5 speech therapists, and 6 other occupations (dentist, medical technologist, biological researcher, practitioner in acupuncture and moxibustion, bonesetter, and certified social worker).

6. The questionnaire should be provided, too.

a. We provided all the questionnaire in the supplemental material from page 2 to 8.

7. Results: The results are expressed as frequencies, the tables should report also the results of Likert scales.

a. We provided the results of Likert scale tables in the supplemental file in Table S2.

8. The paper is interesting, but it reports only descriptive analysis of the healthcare professional’s opinions.

a. As mentioned, this is a descriptive analysis to identify current practices and barriers to implementing ultrasound-based muscle mass assessment. We conducted a web-based questionnaire in the population where sarcopenia evaluation is needed. The practice of muscle mass evaluation varied widely among different occupations. Although the necessity of and interest in ultrasound-based assessment was high among most occupations, only 21% of responders had used the procedure. We found lack of education was the most important barrier that must be overcome to widely implement ultrasound-based assessment. We believe this descriptive analysis contributes to the correct management of sarcopenia.

9. I would suggest to revise the paper and to submit it after again, trying to improve the language and to add more results, if possible.

a. We reanalyzed the data, and revised the manuscript. We added more results in the supplemental file. We conducted English editing to improve the language throughout the manuscript.

---

## [Decision Letter · Decision Letter 1]

11 Aug 2022

PONE-D-21-34367R1Current practice and barriers to implementing ultrasound-based assessment of muscle mass in Japan: A nationwide, web-based cross-sectional studyPLOS ONE

Dear Dr. Nakanishi,

Thank you for submitting your manuscript to PLOS ONE. After careful consideration, we feel that it has merit but does not fully meet PLOS ONE’s publication criteria as it currently stands. Therefore, we invite you to submit a revised version of the manuscript that addresses the points raised during the review process.

This manuscript has been revised based on the comments of a single reviewer. We would be grateful if you could thoroughly assess the manuscript against the journal’s publication criteria as part of your handling of this submission. Unless you acted as the sole reviewer in the first round of review or have concerns that important aspects of the manuscript have not been evaluated in sufficient detail, please try to avoid inviting additional reviewers at this stage.

The reviewers have raised a number of concerns that need attention. They request additional information to be added to the introduction and discussion.

Could you please revise the manuscript to carefully address the concerns raised?

We look forward to receiving your revised manuscript.

Kind regards,

Thomas Phillips, PhD

Staff Editor

PLOS ONE

Journal Requirements:

Reviewers' comments:

Reviewer's Responses to Questions

**Comments to the Author**

1. If the authors have adequately addressed your comments raised in a previous round of review and you feel that this manuscript is now acceptable for publication, you may indicate that here to bypass the “Comments to the Author” section, enter your conflict of interest statement in the “Confidential to Editor” section, and submit your "Accept" recommendation.

Reviewer #1: All comments have been addressed

2. Is the manuscript technically sound, and do the data support the conclusions?

Reviewer #1: Yes

3. Has the statistical analysis been performed appropriately and rigorously? 

Reviewer #1: Yes

4. Have the authors made all data underlying the findings in their manuscript fully available?

Reviewer #1: (No Response)

5. Is the manuscript presented in an intelligible fashion and written in standard English?

Reviewer #1: (No Response)

6. Review Comments to the Author

Reviewer #1: Thank You for the review, according to me the paper was improved. In this version it can be considered for publication. I would suggest correcting of some grammar errors, still present in the text. In the discussion You shold add confrontation of similar papers analysing situation in other countries. In the background I would evidence all the advantages of the method, why it should be used more widely.

7. PLOS authors have the option to publish the peer review history of their article (what does this mean?). If published, this will include your full peer review and any attached files.

Reviewer #1: No

---

## [Author Response · Author response to Decision Letter 1]

29 Aug 2022

1. Thank you for the review, according to me the paper was improved

a. We appreciate reviewer’s comments. We have been encouraged by the reviewer’s positive comment.

2. In this version it can be considered for publication. I would suggest correcting of some grammar errors, still present in the text.

a. We again conducted a professional editing by ENAGO. All changes were underlined. We hope our revised manuscript meets your expectations.

3. In the discussion You should add confrontation of similar papers analyzing situation in other countries.

a. We added confrontation of similar papers analyzing situation in other countries. We added as following.

Consistent with our findings, other surveys on physiotherapists or rheumatologists have also indicated that one of the most common barriers is the lack of training. [18-20]. According to Ellis et al., 76% of non-ultrasound users answered that lack of training was the barrier to using the method [18]. Potter et al. reported that this barrier accounted for 63% of the participants in their questionnaire [19].

Contrary to expectations, approximately 70% of the participants had access to ultrasound equipment, thus indicating that practitioners in Japan can access ultrasound relatively easily. Although access to the equipment acted as the barrier in 56% of non-ultrasound users in a previous study,[18] it may vary from country to country.

Ultrasound-based muscle mass assessment was most frequently conducted by physical therapists (33%), which is consistent with a previous study reporting that 38% of physiotherapists use ultrasound.[18] Moreover, 15% of physicians performed ultrasound-based muscle mass assessment, which is also consistent with a previous study reporting a finding of 17%.[20]

4. In the background I would evidence all the advantages of the method, why it should be used more widely.

a. We added all the advantages of the ultrasound method to explain why it should be used more widely as following.

There are several advantages in the use of ultrasound for muscle mass assessment. Ultrasound is noninvasive and can be employed by various levels of healthcare providers.[16] It can be used continuously to monitor muscle mass because it is available at the bedside. Furthermore, unlike DEXA and BIA, ultrasound can visually assess skeletal muscle mass without being influenced by fluid balance [10].

---

## [Decision Letter · Decision Letter 2]

4 Oct 2022

PONE-D-21-34367R2Current practice and barriers in the implementation of ultrasound-based assessment of muscle mass in Japan: A nationwide, web-based cross-sectional studyPLOS ONE

Dear Dr. Nakanishi,

Thank you for submitting your manuscript to PLOS ONE. After careful consideration, we feel that it has merit but does not fully meet PLOS ONE’s publication criteria as it currently stands. Therefore, we invite you to submit a revised version of the manuscript that addresses the points raised during the review process.

We look forward to receiving your revised manuscript.

Kind regards,

Supat Chupradit, Ph.D., M.Ed., B.Sc.(OT), B.P.A., B.Ed., B.A.

Academic Editor

PLOS ONE

Journal Requirements:

Reviewers' comments:

Reviewer's Responses to Questions

**Comments to the Author**

1. If the authors have adequately addressed your comments raised in a previous round of review and you feel that this manuscript is now acceptable for publication, you may indicate that here to bypass the “Comments to the Author” section, enter your conflict of interest statement in the “Confidential to Editor” section, and submit your "Accept" recommendation.

Reviewer #1: All comments have been addressed

Reviewer #2: All comments have been addressed

Reviewer #3: (No Response)

Reviewer #4: All comments have been addressed

2. Is the manuscript technically sound, and do the data support the conclusions?

Reviewer #1: Yes

Reviewer #2: Yes

Reviewer #3: Yes

Reviewer #4: Yes

3. Has the statistical analysis been performed appropriately and rigorously? 

Reviewer #1: Yes

Reviewer #2: Yes

Reviewer #3: Yes

Reviewer #4: Yes

4. Have the authors made all data underlying the findings in their manuscript fully available?

Reviewer #1: Yes

Reviewer #2: Yes

Reviewer #3: Yes

Reviewer #4: Yes

5. Is the manuscript presented in an intelligible fashion and written in standard English?

Reviewer #1: Yes

Reviewer #2: Yes

Reviewer #3: Yes

Reviewer #4: Yes

6. Review Comments to the Author

Reviewer #1: Dear Authors,

according to my opinion the paper has been improved and can be accepted for pubblication.

Reviewer #2: Overall, this paper is important, interesting, and useful both academically and practically, especially in Japan. It was an extensive and detailed survey that showed some interesting results.

However, I must admit that throughout the paper the authors were only interested in Japan. What authors should also be aware of is that the paper will be published in an international journal and it should provide more benefits than just Japan. At the very least, the Conclusion and Discussion section should reduce some mention of Japan and try to link the findings made in Japan with research in other countries.

Other details that should be improved include;

Abstract should be compacted. Many details, including various stats, can be eliminated.

3-5 keywords need to be added after the Abstract.

Reviewer #3: This manuscript topic: Current practice and barriers in the implementation of ultrasound-based assessment of muscle mass in Japan: A nationwide, web-based cross-sectional study, Well written revise version. I appreciate data analysis and show good results and discussions. Accept

Reviewer #4: Current practice and barriers in the implementation of ultrasound-based assessment of muscle mass in Japan: A nationwide, web-based cross-sectional study, I think this revise version upgrade manuscript, good data show Japan contexts. It's interesting to publish. Accept

7. PLOS authors have the option to publish the peer review history of their article (what does this mean?). If published, this will include your full peer review and any attached files.

Reviewer #1: No

Reviewer #2: **Yes: **Kittisak JERMSITTIPARSERT

Reviewer #3: No

Reviewer #4: No

---

## [Author Response · Author response to Decision Letter 2]

5 Oct 2022

Responses to Reviewer #2:

1. Overall, this paper is important, interesting, and useful both academically and practically, especially in Japan. It was an extensive and detailed survey that showed some interesting results.

a. We appreciate reviewer’s comments. 

2. However, I must admit that throughout the paper the authors were only interested in Japan. What authors should also be aware of is that the paper will be published in an international journal and it should provide more benefits than just Japan. At the very least, the Conclusion and Discussion section should reduce some mention of Japan and try to link the findings made in Japan with research in other countries.

a. We added some statement to provide more benefits out of Japan. We revised discussion and conclusion section as following.

Discussion

Para. 236

According to Ellis et al., 76% of non-ultrasound users answered that lack of training was the barrier to using the method in the international survey [18]. Potter et al. reported that this barrier accounted for 63% of the participants in the United Kingdom [19].

Para. 250

Although access to the equipment acted as the barrier in 56% of non-ultrasound users in a previous study,[18] it may vary from country to country. Indeed, it is a serious problem to ensure ultrasound equipment and its maintenance in developing countries [23].

Para. 261

Although limb circumference can be measured in any country without any specific equipment [24], we need to know limb circumference measurement does not have a good discriminatory power for identifying low skeletal muscle mass because the indirect muscle mass assessment is affected by numerous factors.[25]

Para. 293

Second, this study was conducted in Japan; However, some questionnaire responses were consistent with previous surveys, possibly presenting important suggestions in other countries.

Conclusion

This questionnaire study examined the current practices in and barriers to implementing ultrasound-based muscle mass assessment.

3. Abstract should be compacted. Many details, including various stats, can be eliminated.

a. We compacted the abstract by eliminating many details including various stats as following. The word counts became 243 words from 284 words.

Muscle mass is an important factor for surviving an illness. Ultrasound has gained increased attention as a muscle mass assessment method because of its noninvasiveness and portability. However, data on the frequency of ultrasound-based muscle mass assessment are limited, and there are some barriers to its implementation. Hence, a web-based cross-sectional survey was conducted on healthcare providers in Japan, which comprised four parts: 1) participant characteristics; 2) general muscle mass assessment; 3) ultrasound-based muscle mass assessment; and 4) the necessity of, interest in, and barriers to its implementation. Necessity and interest were assessed using an 11-point Likert scale, whereas barriers were assessed using a 5-point Likert scale, in which “Strongly agree” and “Agree” were counted for the analysis. Of the 1,058 responders, 1,026 participants, comprising 282 physicians, 489 physical therapists, 84 occupational therapists, 120 nurses, and 51 dieticians, were included in the analysis. In total, 93% of the participants were familiar with general muscle mass assessment, and 64% had conducted it. Ultrasound-based muscle mass assessment was performed by 21% of the participants. Necessity and interest scored 7 (6–8) and 8 (7–10), respectively for ultrasound-based muscle mass assessment. The barriers to its implementation included lack of relevant education (84%), limited staff (61%), and absence of fixed protocol (61%). Regardless of the necessity of and interest in ultrasound-based muscle mass assessment, it was only conducted by one-fifth of the healthcare providers, and the most important barrier to its implementation was lack of education.

4. 3-5 keywords need to be added after the Abstract.

a. We added 5 keywords as following.

Keywords

Muscle mass, Sarcopenia, Ultrasound, Dual-energy X-ray absorptiometry, Bioelectrical impedance analysis

---

## [Decision Letter · Decision Letter 3]

17 Oct 2022

Current practice and barriers in the implementation of ultrasound-based assessment of muscle mass in Japan: A nationwide, web-based cross-sectional study

PONE-D-21-34367R3

Dear Dr. Nakanishi,

We’re pleased to inform you that your manuscript has been judged scientifically suitable for publication and will be formally accepted for publication once it meets all outstanding technical requirements.

Kind regards,

Supat Chupradit, Ph.D., M.Ed., B.Sc.(OT), B.P.A., B.Ed., B.A.

Academic Editor

PLOS ONE

Additional Editor Comments (optional):

Reviewers' comments:

Reviewer's Responses to Questions

**Comments to the Author**

1. If the authors have adequately addressed your comments raised in a previous round of review and you feel that this manuscript is now acceptable for publication, you may indicate that here to bypass the “Comments to the Author” section, enter your conflict of interest statement in the “Confidential to Editor” section, and submit your "Accept" recommendation.

Reviewer #1: All comments have been addressed

Reviewer #3: All comments have been addressed

Reviewer #4: All comments have been addressed

2. Is the manuscript technically sound, and do the data support the conclusions?

Reviewer #1: Yes

Reviewer #3: Yes

Reviewer #4: Yes

3. Has the statistical analysis been performed appropriately and rigorously? 

Reviewer #1: Yes

Reviewer #3: Yes

Reviewer #4: Yes

4. Have the authors made all data underlying the findings in their manuscript fully available?

Reviewer #1: No

Reviewer #3: Yes

Reviewer #4: Yes

5. Is the manuscript presented in an intelligible fashion and written in standard English?

Reviewer #1: Yes

Reviewer #3: Yes

Reviewer #4: Yes

6. Review Comments to the Author

Reviewer #1: Dear Authors, the paper was improved, according to me can be accepted for publication. There are some issues related to the english language, but the text il clear.

Reviewer #3: The manuscript has been significantly improved by adding all those changes. I believe that paper is now much more interesting to the future readers and quality of presentation has been raised to a higher level. Therefore, I would like to congratulate the authors on excellent work.

Reviewer #4: Current practice and barriers in the implementation of ultrasound-based assessment of muscle mass in Japan: A nationwide, web-based cross-sectional study. Overall interesting, concise and well written with revision version. I satify this manuscript.

7. PLOS authors have the option to publish the peer review history of their article (what does this mean?). If published, this will include your full peer review and any attached files.

Reviewer #1: No

Reviewer #3: No

Reviewer #4: No

---

## [Editor Report · Acceptance letter]

26 Oct 2022

PONE-D-21-34367R3 

Current practice and barriers in the implementation of ultrasound-based assessment of muscle mass in Japan: A nationwide, web-based cross-sectional study 

Dear Dr. Nakanishi:

I'm pleased to inform you that your manuscript has been deemed suitable for publication in PLOS ONE. Congratulations! Your manuscript is now with our production department. 

Kind regards, 

on behalf of

Assistant Professor Supat Chupradit 

Academic Editor

PLOS ONE